

# Effects of PAHs on meiofauna from three estuaries with different levels of urbanization in the South Atlantic

Renan B. da Silva[1],*, Giovanni A. P. Dos Santos[1],*, Ana Luiza L. de Farias[1], Débora A. A. França[1], Raianne Amorim Cavalcante[1], Eliete Zanardi-Lamardo[2], Jose Roberto Botelho de Souza[1] and Andre M. Esteves[1]

[1] Campus Recife, Center for Biosciences, Department of Zoology, Federal University of Pernambuco, Recife, Pernambuco, Brazil
[2] Campus Recife, Technology and Geoscience Center, Department of Oceanography, Federal University of Pernambuco, Recife, Pernambuco, Brazil
* These authors contributed equally to this work.

## ABSTRACT

Estuarine environments are suggested to be the final receivers of human pollution and are impacted by surrounding urbanization and compounds carried by the river waters that flow from the continent. Polycyclic aromatic hydrocarbons (PAHs) are among the contaminants that can reach estuaries and can directly affect marine conservation, being considered highly deleterious to organisms living in these environments. This research investigated the meiofauna of three estuaries exposed to different levels of urbanization and consequently different levels of PAH concentrations, in order to assess how these compounds and environmental factors affect the distribution, structure and diversity of these interstitial invertebrates. A total of 15 major meiofauna groups were identified, with Nematoda being the dominant taxon (74.64%), followed by Copepoda (9.55%) and Polychaeta (8.56%). It was possible to observe significant differences in all diversity indices studied in the estuaries. With the exception of average density, the diversity indices (richness, Shannon index and evenness) were higher in the reference estuary, Goiana estuarine system (GES). On the other hand, the Timbó estuarine system (TES) had the lowest Shannon index value and richness, while the Capibaribe estuarine system (CES) had the lowest evenness value. The latter two estuaries (TES and CES) presented intermediate and high levels of urbanization, respectively. The ecological quality assessment (EcoQ) in the studied estuaries was classified from Poor to Moderate and the estuary with the lowest demographic density in its surroundings, GES, showed a better ecological quality (Moderate EcoQ). A significant distance-based multivariate linear modelling regression (DistLM) was observed between the environmental variables and the density of the meiobenthic community, where PAHs and pH were the main contributors to organism variation. The sediments were characterized by predominance of very fine sand and silt-clay in the most polluted environments, while the control site environment (GES) was dominated by medium grains. The highest concentrations of PAHs were found in the most urbanized estuaries, and directly affected the structure of the interstitial benthic community. The metrics used in the present study proved to be adequate for assessing the environmental quality of the investigated estuaries.

Corresponding author
Giovanni A. P. Dos Santos,
giopaiva@hotmail.com

## HIGHLIGHTS

1. Meiofauna proved to be a valuable ecological tool for the assessment of anthropic impacts in estuaries with different levels of urbanization and PAH pollution.
2. The evaluation of ecological quality (EcoQ) must be integrated with diversity indices, mainly equitability.
3. A gradient of PAH concentrations from less to more urbanized areas was observed.
4. Benzo[b]fluoranthene and anthracene were among the PAHs that most influenced meiofaunal composition.

## INTRODUCTION

Estuaries are coastal ecosystems with broad environmental variation in salinity, pH and sediment granulometry. Although they are semi-enclosed ecosystems, estuaries are directly connected to the ocean and are influenced by continental drainage and evaporation (*Elliott & Whitfield, 2011*), as well as being influenced by alternating tides (*Jones et al., 2020*). Estuaries are recognized as natural nurseries (*Courrat et al., 2009*), and are also affected by urban expansion and regional processes, such as heat islands caused by the reduction of riparian vegetation, climatic factors, recent sedimentation and local hydrodynamics (*Cui, Zhang & Hua, 2021*; *Hoque et al., 2020*; *Scanes, Scanes & Ross, 2020*).

Although estuarine ecosystems are ecologically important and provide services for biological (*Adams et al., 2006*) and economic maintenance (*Cai & Li, 2011*; *Glaser, 2003*), their proximity to urban areas makes them vulnerable to the entry and chronic deposition of potentially toxic compounds (*Gabriel et al., 2020*; *Han et al., 2020*; *Wang et al., 2021*). Among the pollutants, polycyclic aromatic hydrocarbons (PAHs) are cause for great environmental concern due to their potential toxicity to humans and animals (*Abdel-Shafy & Mansour, 2016*; *Honda & Suzuki, 2020*). These organic compounds contaminate environments mainly as a result of the mishandling of petroleum-based derivatives (*Stogiannidis & Laane, 2015*). Highly industrialized and urbanized areas are more propitious to the release of these pollutants, since PAHs mainly reach coastal and estuarine environments through the release of effluents and untreated domestic sewage, as well as urban runoff (*Domínguez et al., 2010*; *Elmquist, Zencak & Gustafsson, 2007*; *Zakaria et al., 2002*). These multiple sources carry a mix of polycyclic aromatic hydrocarbons, which may accumulate in sediments and cause problems for the surrounding fauna, including mutagenic and carcinogenic effects due to their high toxicity levels (*Engraff et al., 2011*; *United States Environmental Protection Agency, 2018*).

According to their origin, PAHs are classified into two groups: petrogenic, *i.e.*, a direct introduction of petroleum derivatives (crude oil, fuels, lubricants) that flow into affluents and water bodies; or pyrolytic, *i.e.*, PAHs which arise from the partial burning of petroleum derivatives or even from natural organic matter. Both types have different characteristics in

terms of chemical composition and toxicity level (*Abdel-Shafy & Mansour, 2016*; *Zakaria et al., 2002*). Petrogenic sources are mainly comprised of 2–3 aromatic ring compounds and are less toxic, while pyrolytic processes are more toxic and greater molecular weights (4–6 ring-compounds) dominate (*Boehm, 2005*).

Contamination of estuaries with PAHs can potentially create disturbances in benthic assemblages, including the benthic meiofauna (*Schratzberger & Ingels, 2018*). It is important to highlight that the conservation of marine ecosystems may be achieved efficiently based on these tiny animals (*Balsamo et al., 2012*). Some studies suggest that investigating changes that occur in animals at the bottom of the food chain, rather than between more charismatic and larger animals, allows for a more efficient monitoring and timely conservation responses (*Ingels et al., 2021*). Meiobenthic organisms are widely used as indicators of ecological impacts and for biomonitoring, because (I) they share a close relationship with the sediment, (II) do not present larval dispersion and (III) are sensitive to environmental changes (*Hyland et al., 2005*; *Pusceddu et al., 2007*). In addition to diversity indices, the use of meiofauna allows for the application of the ecological quality status (EcoQ), an index widely used both in studies of open habitats (*Chen, 2018*) and semi-enclosed environments (*Semprucci, Balsamo & Sandulli, 2016*). PAHs are commonly found in areas impacted by oil residue, however there is ambiguity as to their deleterious effects on the benthic community, since reductions in richness with increases in abundance have been previously documented (*Baguley et al., 2015*; *Erstfeld & Snow-Ashbrook, 1999*; *Montagna et al., 2013*; *Zeppilli et al., 2015*). As such, in addition to evaluating the effects of total PAHs on meiofauna and their diversity indices, including EcoQ, we investigated the effects of each PAH concentration, individually.

The aim of the present study was to characterize the meiofaunal groups inhabiting three tropical estuaries with different levels of urbanization, and to relate taxon composition, richness and diversity to concentrations of polycyclic aromatic hydrocarbons (PAHs) and environmental variables (pH, organic matter, dissolved oxygen, temperature, salinity and granulometry). Considering that meiofauna has been shown to be a good indicator of different global impacts and changes, the hypothesis is that (i) environmental variables and (ii) polycyclic aromatic hydrocarbon concentrations significantly correlate with the spatial variation of meiofaunal structure.

## MATERIALS AND METHODS

### Study area

The studied areas were the tropical estuaries of Goiana, Timbó and Capibaribe rivers, located on the East coast of South America, Northeastern Brazil (Fig. 1). All of the study areas are urbanized to different degrees. The Goiana estuarine system (GES) is located at 7°32′43.2″S, 34°51′50.2″W. A study by the state environmental agency considered this estuary as poorly urbanized (*CPRH, 2005*), although it receives PAHs mainly from the burning of sugarcane straw (*Arruda-Santos et al., 2018*). Some studies performed in this area showed low PAH contamination (*Arruda-Santos et al., 2018*), and it was previously classified as undisturbed, using the biotic index (AMBI) (*Nunes de Souza et al., 2021*). The Timbó estuarine system (TES) is located at 7°53′45.8″S, 34°51′35.9″W, and has an

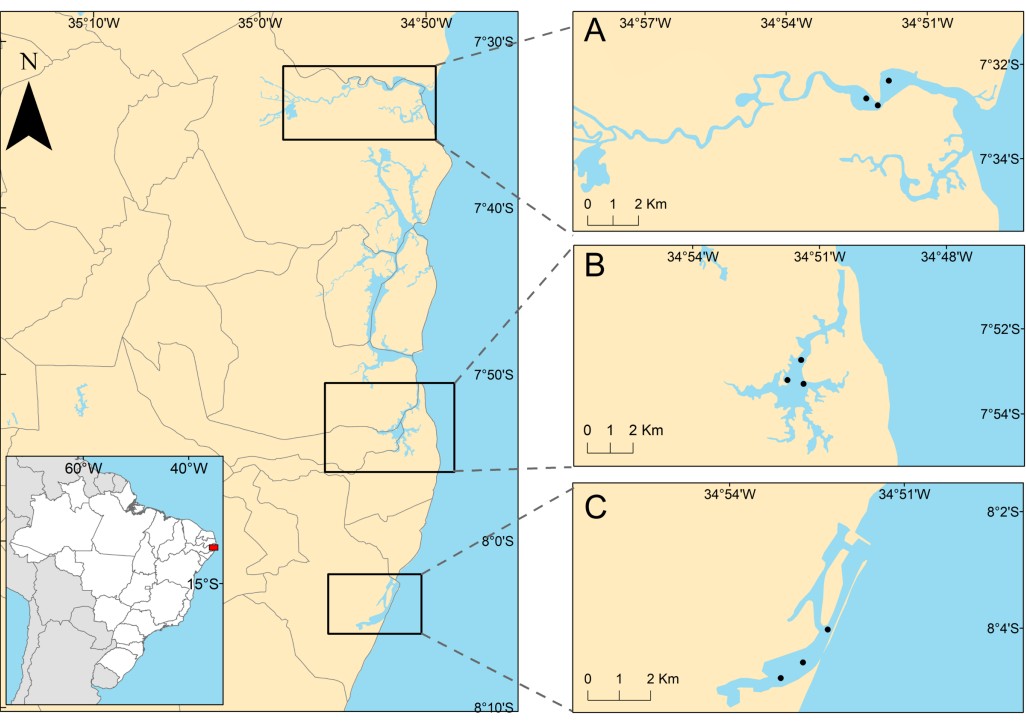

**Figure 1 Positions of sampling stations in the estuaries.** Positions of sampling stations in the study estuaries, located on the northeastern coast of Brazil. (A) Less urbanized area three Goiana estuarine system (GES); (B) intermediate urbanization area—Timbó estuarine system (TES); (C) more urbanized are—Capibaribe estuarine system (CES).

intermediate level of urbanization (*CPRH, 2005*), mainly receiving pollution from untreated and industrial sewage (*Noronha, da Silva & Duarte, 2011*). The AMBI index identified TES as slightly (*Valença & Santos, 2012*) to moderately disturbed (*Nunes de Souza et al., 2021*), suggesting an increasing degradation of this estuary. The Capibaribe estuarine system (CES) is located at $8°04'03''$S and $34°52'16''$W in the innermost part of Recife's Port and is a highly urbanized area (*CPRH, 2005*) (see discussion section). It is formed by the confluence of the Tejipió, Jordão and Pina rivers and the Southern arm of the Capibaribe river. Based on the AZTI biotic index, this estuary was classified as moderate to highly disturbed (*Valença & Santos, 2012*; *Nunes de Souza et al., 2021*). Several studies reported that sediments from the CES are contaminated by PAHs, aliphatic hydrocarbons, tributyltin (antifouling compounds), and metals (*Macedo et al., 2007*; *Maciel et al., 2015*, *2016*, *2018*).

## Sampling

The samples were obtained in January and February 2016, during the summer (dry) period. Sample collection was carried out during this season in order to avoid the period of high rainfall that leads to the enrichment of other pollutants, fertilizers and pesticides, which would interfere with the benthic fauna and the assessment of the effects of PAHs (*Boonyatumanond et al., 2006*; *Zakaria et al., 2002*; *Zhang et al., 2017*). The estuaries were sampled at three points, and four replicas were obtained at each point using a Van Veen

(nine liters) bottom sampler. Meiofaunal samples were obtained with the aid of a 5 cm tall cylinder with an internal diameter of 3.6 cm (area of 10 cm$^2$). Samples were preserved with 4% buffered formaldehyde (*Giere, 2009*).

For the analysis of sediment chemical (PAHs, Total Organic Content) and physical parameters (grain size), sediment fractions were separated within each replicate. In order to avoid contamination of the fraction intended for PAH analyses, a stainless-steel spatula was used to scoop out the surface of the sediment (~2 cm) and a sterile aluminum container was used to store each sample frozen until PAH processing and reading was performed. The samples were homogenized before the chromatographic analysis and PAH characterization, in order to avoid punctual variation and chemical agglomeration. Salinity, temperature, dissolved oxygen and pH were measured using a JFE Advantech type CTD probe, Rinko Profiler model.

## Treatment of biological samples in the laboratory

To separate the meiofauna from the sediment, running water and sieves (300 and 45 µm coupled meshes, respectively) were used. The sediment remaining in the 45 µm mesh underwent a process of ten manual elutriations and the supernatant from each elutriation was removed and fixed with 4% buffered formalin (*Giere, 2009*). Meiofauna was identified with the aid of a stereomicroscope at the level of major groups (*Higgins & Thiel, 1988*).

## Analyzes of the sediment variables

Organic matter was calculated from weight loss following ignition at 450 °C for 5 h (*Danovaro, 2009*). Granulometry was determined following *Suguio (1973)*, using the wet method to sieve and separate the silt-clay fraction. The remaining sediment was sieved in a shaker after being dried and weighed and fractionated through sieves with openings of sizes 2 mm to 0.062 mm. The methods for analyzing PAH concentrations are described in *Nunes de Souza et al. (2021)*. Briefly, the samples were analyzed by gas chromatography (model 7820A; GC—Agilent Technologies, Santa Clara, CA, USA) coupled with mass spectrometry (model 5975C; MS—Agilent Technologies, Santa Clara, CA, USA), in the selected ion monitoring mode (SIM). The values presented here refer to the sum of the PAHs ($\sum$PAH).

## Data analysis

Meiofauna density data were transformed into the fourth root and the similarity matrix was calculated using the Bray-Curtis index. In order to visualize the similarity patterns they were ordered using a non-metric multidimensional scaling technique (nMDS). To test the significance of the visualized patterns, a permutational ANOVA (PERMANOVA) was applied. A PERMDISP analysis was used to test homogeneity among data, and the estuary areas (GES, TES and CES) were applied as factor. All abiotic data were transformed (Log(V + 1)) and normalized before being used in correlation analyses.

To assess estuarine diversity, the following indices were calculated: Shannon Wiener (H′), Pielou (J) and richness values (S) (*Anderson, Gorley & Clarke, 2008*). To analyze these ecological indices of meiofauna responses to environmental factors and PAH

concentrations, the Distance-based multivariate linear modelling regression (DistLM) was applied. The dbRDA was performed to achieve the ordination and visualization of fitted models (such as from DistLM), and ploted vectors in graphics were generated with Spearman's rank correlation.

The environmental quality status (EcoQ) was obtained for each station of the estuaries, by considering the total number (richness) of meiobenthic taxa as proposed by *Danovaro et al. (2004)*, modified according to European Water Framework Directive (WFD), that define environmental quality in the following classes: group richness ≤4, bad; between five and seven groups, poor; between eight and 11 groups, moderate; between 12 and 15 groups, good; ≥16 groups, high.

Multivariate analyzes were performed using the software: PRIMER v6 with the addition of the PERMANOVA+ package (*Gorley & Clarke, 2008*).

## RESULTS

### Environmental variables

Estuary salinity ranged from 23 to 36, and the average temperature was 29.2 ± 0.1 °C (average ± SE). Organic matter (OM) varied greatly between estuaries, ranging from 1.37% to 16.28% (pseudo-F = 21.61; $p = 0.014$) with a significantly lower amount of OM at Goiana estuarine system (GES) ($p < 0.045$). The pH ranged from 5.79 to 8.50, where significantly more acidic measurements were recorded for the Capibaribe estuarine system (CES), differing from the others ($p < 0.0003$). Interestingly, dissolved oxygen was at least three times higher at CES, differing in the pairwise comparison that was registered at GES and at Timbó estuarine system (TES) ($p < 0.02$) (Table SA1).

Regarding PAHs, a total of 17 different compounds were identified, of which 16 are listed as harmful to health by the United States Environmental Protection Agency (US EPA) (*Keith, 2015*). Concentrations ranged from 0.55 ± 0.67 (average ± SE) in GES, to 674.81 ± 331.31 in CES (Table SA1). Only three PAHs were detected in the GES sediments: 2-methyl naphthalene, fluorene and phenanthrene. Most PAHs were common to CES and TES, with the exception of acenaphthene, that was detected in CES sediments. The highest individual concentrations of PAHs were reported for fluoranthene, indeno[1,2,3-cd] pyrene, pyrene, benzo[a]pyrene, and benzo[b]fluoranthene, which together accounted for 56.2% of the total PAH concentration.

The granulometry in the estuaries ranged from gravel to silt-clay, where the average sand fraction was more abundant in the estuary less impacted by PAHs, while the most polluted areas were dominated by silt-clay. The comparison of sediment fractions between areas showed that very fine sand was significantly lower in the GES, whereas the silt-clay fraction in the CES was significantly higher compared to GES (Table SA2). The sedimentary matrix was classified as poorly selected in the most polluted estuaries (CES and TES). On the other hand, in the reference estuary, only one station was classified as poorly selected, and the other two stations were classified as moderately selected and dominated by medium grains (*Nunes de Souza et al., 2021*).

## Meiofauna

A total of 22,152 individuals were identified, distributed across 15 major meiofaunal groups: Nematoda, Copepoda, Rotifera, Turbellaria, Tardigrada, Gastrotricha, Ostracoda, Halacaroidea, Oligochaeta, Cnidaria, Polychaeta, Amphipoda, Sipuncula, Kinorhyncha and Priapulida, added to Nauplius (crustaceans' larvae). Three of these were exclusive to GES (Sipuncula, Kinorhyncha, Priapulida), and one was exclusive to TES (Amphipoda). Richness varied significantly across estuaries (pseudo-F = 13,537; $p$ = 0.0001), ranging from three at TES to 11 at GES. On average, the richness at GES was 8.75, followed by 7.83 ± 0.46 in the CES and 5.33 ± 0.55 in the TES, where the latter presented the lowest average and differed from the others in the pairwise comparison ($p$ < 0.0014).

Meiofauna density (ind./10 cm$^2$) varied significantly from one estuary to another (pseudo-F = 9.7861; $p$ = 0.0002), with significantly higher values in the CES than in the other estuaries ($p$ < 0.0065). Average densities (±SE) were 1,069.2 ± 194.4 at CES; 452.2 ± 74.5 at GES and 324.7 ± 91.8 at TES. Among the taxa, Nematoda were the most abundant group, and when combined with Copepoda and Polychaeta they accounted for more than 90% of the identified organisms. The high-density value within the CES was mainly due to the presence of Nematoda and Polychaeta, whereas Nematoda and Copepoda were the most abundant groups in the GES and TES (Fig. 2). Although appearing in smaller percentages, less than 2% (Fig. SA1), it is worth noting the presence of Tardigrada and Gastrotricha in the GES, and the increase of Ostracoda in the CES, as well as, Rotifera in both TES and CES (Fig. 3).

The PERMANOVA analysis and the pairwise comparison revealed that community structure differed significantly between estuaries (Table 1). The nMDS showed a greater association between CES and TES compared to the GES stations (Fig. 4). The Spearman's correlation applied to the nMDS (vectors) indicated that Gastrotricha, Turbellaria and Tardigrada, respectively, presented most dominance and positively correlation with GES were also negatively correlated with TES. The Rotifera and Polychaeta taxa correlated positively with both CES and TES.

Shannon's diversity (H) ranged from 2.31 at GES to 1.02 at TES. This index differed between estuaries (Table SA3), showing higher values at GES compared to all other locations. Evenness varied significantly from one estuary to another, where the value of this index was significantly lower at CES compared to GES and TES (Table SA3). Based on meiofaunal richness (S), the ecological quality status (EcoQ) of all three TES station and CES station 2 were classified as Poor (S = 1–4), and all of GES station; CES station 1 and 3 as Moderate (S = 8–11) (Table SA4).

## Correlation between meiofauna and environmental variables

The multivariated DistLM-Best analysis for environmental data explained 60.94% of the meiofauna community variation (DistLM—Best) (Table SA5). Among the environmental data tested, salinity ($p$ = 0.5) and some sediment fractions did not significantly influence community variation (coarse sand: $p$ = 0.46; fine sand: $p$ = 0.28). It was possible to highlight that the sum of the PAHs and pH contributed to more than 39% of the total meiofauna variation (Table SA5). The dbRDA showed a clear separation between the least

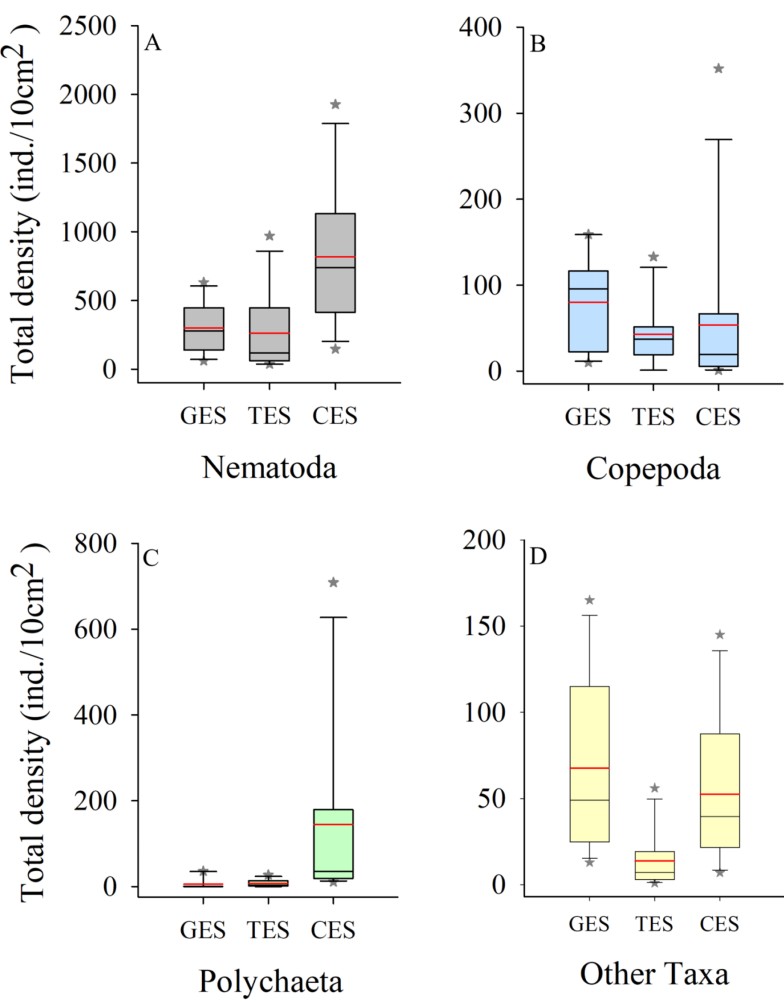

**Figure 2 Density of the most abundant groups.** Density of the most abundant groups registered in each estuary. Median (solid black line), Average (in red), whisker boxes represent upper/lower quartiles. Vertical lines extending from each box represent the minimum and maximum value, an asterisk (*) indicated outliers that outranged the box limits. (A) Nematoda density, (B) Copepoda density, (C) Polychaeta density and (D) Density of the other groups recorded in the estuaries. GES, Goiana estuarine system; TES, Timbó estuarine system; CES, Capibaribe estuarine system.

polluted area (GES) and the other two areas (CES and TES), as well as the relevance of PAHs and pH for the distinction between different groups (Fig. 5).

The Spearman rank of correlation for individual and total PAHs concentration significantly correlate with the reduction of Tardigrada, Gastrotricha, Halacaroidea, and Sipuncula ($p < 0.01$), but favor the increase of Nematoda, Polychaeta, Rotifera, Ostracoda e Cnidaria ($p < 0.01$). In contrast, higher pH values correlated positively and significantly with Tardigrada ($p = 0.002$) and Gastrotricha ($p = 0.04$) abundance. However, among the most abundant taxa this relationship was inverted for Nematoda and Polychaeta, since they showed a direct and positive relationship with the high acidity registered in the more urbanized estuary (pH: 5.81 ± 0.02) (Tables SA6 and SA7).

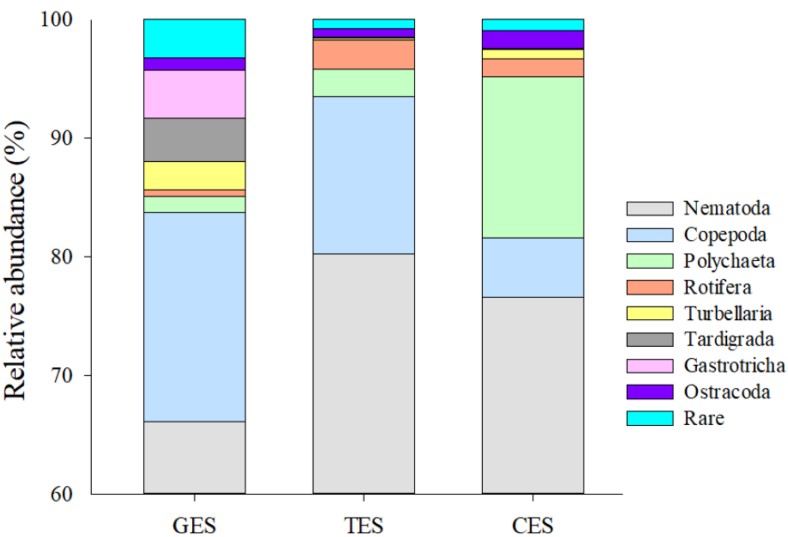

**Figure 3** **Meiofauna relative abundance.** The relative abundance (%) of meiofaunal groups distributed in the three Estuaries. GES, Goiana estuarine system; TES, Timbó estuarine system; CES, Capibaribe estuarine system. Note that the Y axis started with 60% to give a better representation of the less abundante meiofaunal groups.               

**Table 1 Comparison of estuarine meiofauna community structure, as well as pairwise comparison.**

| PERMANOVA | df | MS | Pseudo-F | P (perm) |
|---|---|---|---|---|
| Estuary | 2 | 6,748.9 | 16.251 | **0.0001** |
| Res | 33 | 460.53 | | |
| Total | 35 | | | |
| PERMIDISP | 2 | | 1.6914 | 0.2831 |
| | | | | PAIR-WISE |
| Groups | | | t | P (perm) |
| GES, TES | | | 4.1003 | **0.0001** |
| GES, CES | | | 4.9839 | **0.0001** |
| TES, CES | | | 3.1187 | **0.0003** |

Note:
Results of PERMANOVA, PERMIDISP and PAIR-WISE tests on the structure of the meiofauna communities in the study estuaries. The analysis factor was the area (Estuary). Values of P (perm) < 0.05 are in bold. GES, Goiana estuarine system; TES, Timbó estuarine system; CES, Capibaribe estuarine system; df, degrees of freedom; MS, mean squares; Res, residual.

The multivariate DistLM analysis for individual PAH concentrations explained 59.40% of the total faunal variation. All PAHs significantly influenced organism variation in the estuaries, with benzo[b]fluoranthene and anthracene best explaining the variation, accounting for 46.44% of the total variation (Table SA8). The first two axes of the dbRDA generated in this analysis explained about 86.3% of the PAH distribution within the estuaries. From the vectors in the dbRDA, it is possible to observe a positive correlation between PAHs and the more urbanized estuaries (CES and TES), whereas the correlations with the less urbanized estuary (GES) were all negative (Fig. SA2).
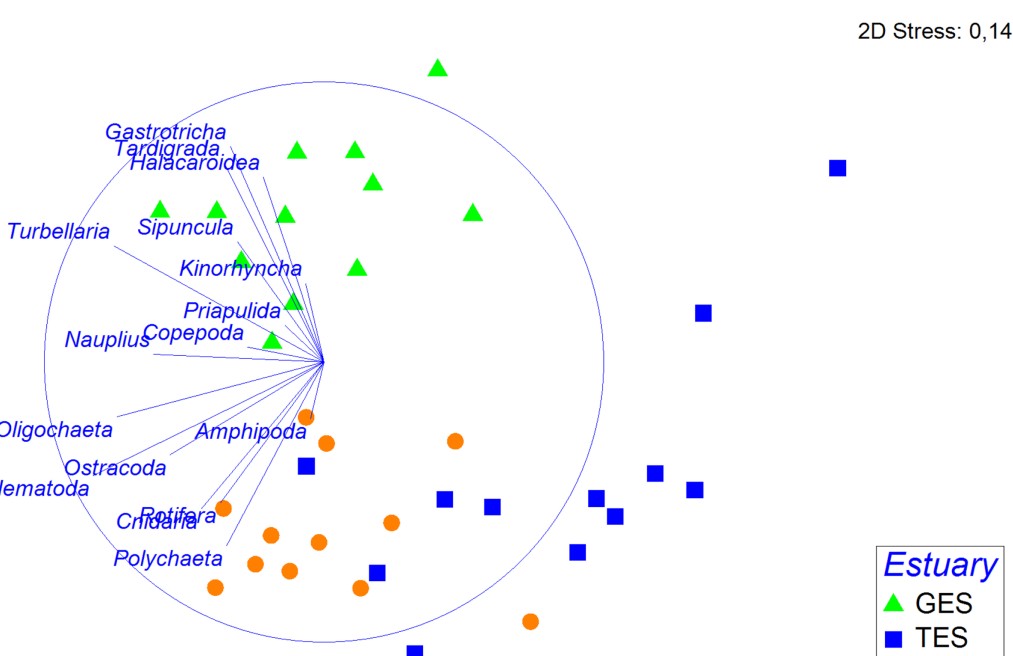

**Figure 4 Non-metric multidimensional scaling (nMDS) based on the density of meiofauna groups.** Non-metric multidimensional scaling (nMDS) based on meiofauna group densities (standardized on the 4th root, using Bray—Curtis), with their vector (strength and direction of effect of the variable on the ordination plot) in the three estuaries tested: GES, Goiana estuarine system; TES, Timbó estuarine system; CES, Capibaribe estuarine system.

Some of the environmental variables correlated significantly with total meiofaunal density and diversity indices. Organism density correlated significantly and positively with total PAHs, dissolved oxygen, temperature and silt-clay, but negatively with the sandy and coarse fraction of the sediment. The Shannon diversity index correlated negatively and significantly with organic matter and very fine and very coarse sand. These sediment fractions were negatively and significantly correlated with richness. Evenness was significantly negatively affected by ∑ PAH, dissolved oxygen, organic matter, as well as very fine sand and silt-clay (see Table SA9). When carried the correlation between the individual PAHs concentration and diversity indices, both density and equitability were affected negatively and significantly (Table SA10).

# DISCUSSION

## Origin of PAH pollution in estuaries

The variation and distribution of organisms in marine ecosystems is linked to the environmental factors of each location (*Coull, 1999*). Estuarine ecosystems are characterized by oscillations in environmental parameters (*i.e.*, salinity, organic matter, granulometry, *etc.*), making them unstable in time and space. These oscillations are associated with the characteristics of the areas through which the converging rivers pass during their course, forming this ecosystem. Common impacts on tributary courses include waste derived from urbanization, agriculture and industrialization in surrounding

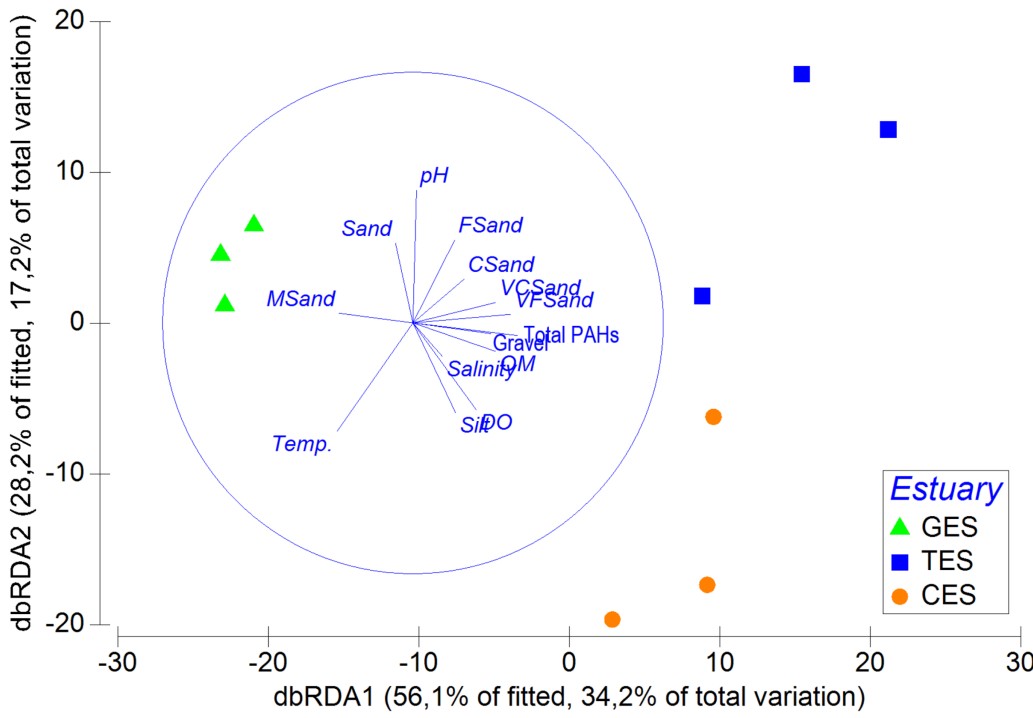

**Figure 5 Distance-based redundancy analysis (dbRDA), correlation between meiofauna assemblage and environmental data.** Distance-based redundancy analysis (dbRDA), correlation between meiofauna assemblage and environmental data (including $\sum$ PAHs) with their vector (strength and direction of effect of the variable on the ordination plot) in the three estuaries tested: GES, Goiana estuarine system; TES, Timbó estuarine system; CES, Capibaribe.

areas, as well as the misuse of their banks resulting in leaching and silting processes (*Balsamo et al., 2012*).

PAHs carried by rivers may be deposited into estuarine sediment mainly through association with salts, sediment grains and organic matter, which can directly affect the resident biocenosis (*Maciel et al., 2015*). These events raise concerns, especially in more populated areas due to the higher incidence and recurrence of impacts and human pollution affecting estuaries. The study area used GES as a reference site due to the lower level of urbanization (*IBGE, 2020*) has a population density of 150.72 inhab/km$^2$ and presented considerably lower concentrations of PAHs compared to the other sites. Population densities in estuaries with medium and high levels of urbanization showed PAH values that were 22 and 46 times higher, respectively.

Although the impacts related to urbanization in the GES were lower, residues from activities such as aquaculture and practices of burning of sugar cane are carried out throughout this estuary basin. Such activities have already been identified as sources of PAH introduction into the Goiana estuarine system (GES), mainly by atmospheric deposition, although in low concentrations (*Arruda-Santos et al., 2018*). Estuaries such as Ohiwa, New Zealand, experience human impacts similar to those observed in the more conserved area of this study (*i.e.*, rural pastures and less inhabited areas), and presented similar total PAH concentrations (Goiana: 1.7 ng/g$^{-1}$; Ohiwa: 3.0 ng/g$^{-1}$) (*Hack et al.,*

*2007*). Based on the Canadian classification of the Sediment Quality Guidelines (SQC), high sediment quality was recorded in the GES, since the PAH concentrations observed in this estuary were lower than the threshold proposed for the rare category (REL) (*ECM, 2007*; *He et al., 2014*). However, these concentrations indicate the entry of PAHs into this area, highlighting the importance of regular monitoring of this estuarine system.

The Timbó estuarine system (TES), which presents an intermediate level of urbanization, experiences a lack of sanitation and receives untreated effluents, generating an increase in organic matter particles in this environment (*Santana, Fernandes & Machado, 2017*). This increase becomes a negative factor for estuary health, since it is usually associated with and facilitates the deposition and accumulation of PAHs and other contaminants in the sediment (*Medeiros & Caruso Bícego, 2004*). Consequently, there is a loss of biodiversity, disruption of the food web, as well as a decrease in ecosystem services provided by benthic organisms (*Gambi et al., 2020*; *Louati et al., 2013*). Most PAHs in TES sediments had individual concentrations that, according to the SQC, would have a low probability of causing negative effects on benthic fauna (*ECM, 2007*). However, the presence of benzo[a]anthracene, dibenz[a,h]anthracene, and fluoranthene compounds raise concerns, since their individual concentrations are close to the threshold effect level (TEL) for benthic organisms (*ECM, 2007*; *He et al., 2014*).

Among the studied sites, the Capibaribe estuarine system (CES) had the highest PAH concentration. As seen in previous studies, this area presents characteristics that facilitate the retention of organic matter and PAHs (*Araújo, Mineiro & de Cantalice, 2011*; *Maciel et al., 2015*). The waters of this estuary are connected to Recife's Port, which receives numerous vessels annually (*Port of Recife, 2021*), becoming a vector area for the release of PAHs. The PAH concentrations observed in the CES are similar to those analyzed in urban and highly industrialized estuaries, such as the Yangtzen River in eastern China (*Li et al., 2012*), as well as in port areas in the Adriatic Sea (*Baldrighi et al., 2019*). The meiofaunal abundances reported in the aforementioned port area were similar to the results of this study, where the abundance of sensitive taxa (Tardigrada and Gastrotricha) decreased due to the high PAH values, while the abundances of more tolerant taxa such as Nematoda and Polychaeta increased (*Dal Zotto et al., 2016*; *Moreno et al., 2011*; *Pusceddu et al., 2007*). It was possible to observe that the concentrations of six of the 17 PAHs were high enough to exceed the threshold effect level (TEL) in at least one station of the CES (PAHs: acenaphthylene, phenanthrene, fluoranthene, benzo[a]anthracene, benzo[a]pyrene, and dibenz[a,h]anthracene). The levels of PAHs recorded at CES were even more alarming than the findings for the TES, since more types of PAHs have concentrations that exceed the threshold to affect fauna (*ECM, 2007*; *He et al., 2014*).

It is worth noting that estuarine environments are subject, not only to PAHs, but to several other contaminants. In fact, previous benthic studies from these urbanized estuaries also reported heavy metal contamination (*Noronha, da Silva & Duarte, 2011*; *Silva et al., 2011*). These contaminants can directly affect the abundance and diversity of meiofauna (*Moens et al., 2014*). Additionally, agricultural fertilizer can be observed (*Noriega et al., 2019*) which can cause eutrophication and anoxia due to bacterial proliferation (*Carriço et al., 2013*). Furthermore the input of tributyltin (*Maciel et al., 2018*)

can interrupt faunal reproductive processes and cause juvenile deaths (*Schratzberger et al., 2002*).

## Ecological quality assessment using EcoQ and variation of meiofauna diversity

The ecological quality classification of the studied estuaries ranged from poor to moderate, with a prevalence of moderate in the estuary with lower PAH concentrations (GES). The estuary with intermediate PAH concentrations (TES) received the poorest classification. This finding suggests that some other stressors are likely altering the quality of these estuaries. Similar richness results observed in the TES were reported in the Venice lagoon in Italy, which presented EcoQ results varying between Bad and Poor (*Pusceddu et al., 2007*). As in Venice (*Zonta et al., 2007*), the main factors highlighted for the bad EcoQ evaluation in the TES were: sewage entry, as well as residues from the industrial district present in the surrounding area, which has released trace elements such as heavy metal and PAHs into the estuary for decades (*Semprucci, Balsamo & Sandulli, 2016*; *Noronha, da Silva & Duarte, 2011*). Such impacts corroborate results reported for the most polluted estuary of this study (CES), which justifies the low abundance (or absence) of sensitive organisms, as well as the low environmental indices' values (*i.e.*, richness, density and Shannon index) and poor EcoQ at the CES stations.

The lower level of urbanization and anthropogenic input are not the only factors responsible for superior ecological quality in the reference estuary. The sedimentary structure, composed mainly of medium grains and organic matter with concentrations 3 to 6 times lower than the other estuaries, favors a lower PAH accumulation in the reference area (*Evans, Gill & Robotham, 1990*; *He et al., 2014*). In fact, the concentration in GES was at least 250 and 1,225 times lower compared to TES and CES, respectively. Additionally, many of the taxa found in polluted estuaries were extremely rare, with low equitability. These organisms did not account for even 0.5% of the total abundance, both in the estuary with an intermediate PAH concentration (Oligochaeta, Turbellaria, Nauplius, Halacaroidea, Amphipoda, Tardigrada, Cnidaria), and in the estuary with the greatest impact (Oligochaeta, Cnidaria, Nauplius, Gastrotricha, Halacaroidea) (*Zeppilli et al., 2015*).

Although the EcoQ of the CES is comparable to that of the least contaminated area (GES), the distribution of taxa in the CES was significantly less equitable compared to the other estuaries. As EcoQ is closely associated with richness, it is worth noting that the equitability that sustains this richness can be used as a parameter to establish the reliability and strength of this status. Thus, even if the EcoQ level is higher because an organism that appears at random, the oscillation of this taxa in the other samples can be seen as a weakening of the EcoQ status.

The fauna distribution was negatively and significantly affected by the high concentrations of organic matter, very fine sand and silt-clay, in addition to the high PAH concentrations present in estuaries located near urban areas. The decrease in environmental indices, such as Shannon diversity, richness, and evenness, are indicative of environmental stress in studies where benthic fauna has been considered as a proxy for

environmental quality status evaluations (*Damasio et al., 2020*; *Janakiraman et al., 2017*; *Warwick et al., 1990*).

Therefore, the aforementioned lower environmental indices' values, further enforce the ecological damage in areas impacted by higher concentrations of PAHs and probably other contaminants (that were not evaluated in this study). These lower environmental quality indices' values, when compared to the reference area, demonstrate that meiofauna are an adequate tool to detect early changes in impacted areas, exhibiting especially detailed responses to pollutants (*Semprucci, Balsamo & Sandulli, 2016*; *Schratzberger & Ingels, 2018*; *Ingels et al., 2021*). Meiofauna showed a better correlation with PAH compounds than macrofauna, presenting a direct correlation between diversity and pollution. On the other hand, macrofauna was more diverse in polluted estuaries, suggesting that these organisms were exposed to an intermediate disruption, according to Multimetric indices such as AMBI (A Marine Biotic Index) (*Borja, Chust & Muxika, 2019*; *Nunes de Souza et al., 2021*).

## Environmental factors shaping meiofauna structure in three different urbanization lavel stuaries

Conservation strategies aim to protect habitats, in addition to understanding the loss of fauna and the ecosystem services they provide to the environment (*Ingels et al., 2021*). The changes that marine environments experience due to anthropogenic actions, affect meiofauna, causing decreases in diversity and richness and mainly result in the loss of more sensitive and rare taxa. When this occurs, individuals with broader niches can proliferate, as they are more resistant or even opportunistic (*Pusceddu et al., 2007*; *Supp & Ernest, 2014*). Nematoda (76.56%) stood out in the studied estuary with the highest level of pollution, followed by Polychaeta (13.53%). On the other hand, the density of Copepoda, which respond more sensitively to areas impacted by anthropogenic action (*Soetaert et al., 1995*), was 15 times lower than that of Nematoda.

Nematoda dominance in the most polluted estuary did not deviate from previously observed patterns in coastal marine environments in southern Italy and northern Iran (*Bertocci et al., 2019*; *Zarghami et al., 2019*), in estuaries present in southeastern India and northern Taiwan (*Cai & Li, 2011*; *Chinnadurai & Fernando, 2007*), and in the deep sea in the Gulf of Mexico (*Baguley et al., 2006*) and Espírito Santo basin in southeast of Bazil (*dos Santos et al., 2020*). Nematodes are commonly the most abundant taxon in areas with greater anthropogenic activity and, notably, with high concentrations of organic matter and PAHs (*Zeppilli et al., 2015*). This is due to the opportunistic characteristics that some colonizing Nematoda genera present, together with their diet composed of bacteria that proliferate from the decomposition of organic content (*Bongers & Ferris, 1999*; *dos Santos et al., 2009*, *2008*; *Schratzberger & Ingels, 2018*).

The fauna pattern observed in estuaries exposed to lower (GES) and intermediate (TES) concentrations of PAHs presented similar meiofauna proportions in natural estuarine environments classified by *Coull (1999)*, where Nematoda dominated with 0–90% and Copepoda accounted for 0–40% of the total abundance. In the GES and TES, Nematoda dominated with 66.09% and 80.17%, respectively, followed by Copepoda with 17.66% and

13.27%. The relative abundance of Copepoda in the estuary least impacted by PAHs was higher compared to the other study areas. Furthermore, the presence of taxa, such as Tardigrada and Gastrotricha, in polluted estuaries was much lower and even null compared to that observed at GES. The decrease in sensitive taxa in more urbanized estuaries confirmed the negative effect of pollution from constant anthropogenic activities, such as effluent and sewage discharge, as well as port and industrial activities, which are all factors that favor the entry of PAHs and other contaminants into estuaries (*van Damme, Heip & Willems, 1984*; *Maciel et al., 2015*; *Zeppilli et al., 2015*).

Taxa such as Tuberllaria, Tardigrada, Gastrotricha, Halacaroidea, Sipuncula and the Equitability index'(J'), correlated negatively with the fractions of very fine sediment and silt-clay, mainly present in the most polluted estuaries. On the other hand, in the less polluted area, medium and moderately selected grains predominated, resulting in larger interstitial spaces and favoring meiofauna diversity. Tardigrada and Gastrotricha, which were abundant in the reference estuary, are adapted to survive in ample interstitial spaces, whereas in areas where the sediment is muddy, such as in CES and TES, these groups try to adapt to epibiont life and become less diverse and abundant (*Giere, 2009*).

Larger proportions of fine grains and silt-clay were registered in the estuaries with higher PAH concentrations and the relationship between these factors has also been reported by other authors (*Tolosa, Mesa-Albernas & Alonso-Hernandez, 2009*; *Maciel et al., 2015*; *Egres et al., 2019*; *Wang et al., 2021*; *Zanardi-Lamardo et al., 2019*; *Losi et al., 2021*). Most fine grains exhibit a greater amount of organic matter, contributing to PAH accumulation and, consequently directly affects meiofauna, either through ingestion or direct contact with contaminants (*Arruda-Santos et al., 2018*; *Stogiannidis & Laane, 2015*; *Tremblay et al., 2005*). Although PAHs best explained fauna distribution among the studied parameters, grain size also played an important role in the heterogeneity of the meiofauna groups, which is one of the main factors influencing meiofauna abundance, distribution and diversity (*He et al., 2014*).

The sedimentary characteristics presented by CES favored the presence of Polychaeta, the high abundance of this group has already been reported in another anthropized estuary, located in Mondego (Portugal), where Polychaeta were the second most abundant group, following Nematoda (*Alves et al., 2013*). While the results of this manuscript suggest positive and significant correlations of Polychaeta meiofauna with $\sum$ PAH, the same was not observed for the coast of Galicia, where no correlations were found between Polychaeta density, environmental parameters, and PAH pollution (*Veiga, Rubal & Besteiro, 2009*). When exposed to acute impacts, such as the Deepwater Horizon accident, there was a decrease in Polychaeta meiofauna at points close to the accident, however Polychaeta macrofauna families (*eg.*, Spionidae, Capitellidae, Maldanidae) were dominant at the spill site (*Baguley et al., 2015*; *Jewett, Thomas & Blanchardl, 1999*; *Washburn, Rhodes & Montagna, 2016*).

Although most polychaeta macrofauna produce temporary juvenile meiofaunal larvae, this apparently did not affect polychaeta abundance in the previously mentioned studies. The estuary most polluted by PAHs (CES) housed more than 90% of the total Polychaeta meiofauna recorded in the present study, among which there was a slight predominance of

the Spionidae family (data not shown). The correlation between $\sum$ PAH and polychaetes was also observed in macrofauna associations, with polychaetes representing more than 70% of abundance in TES and CES, and the Spionidae family representing 73% of the total abundance of this estuary (*Nunes de Souza et al., 2021*). The bodies of adult Spionidae individuals range from 1.3 to 150 mm and their larvae temporarily belong to meiofauna (*Radashevsky, 2012*). This family is commonly associated with environmental disturbances in marine environments, as its abundance increases when exposed to environmental stress (*Dean, 2008*). Additionally, this family presents opportunistic species with rapid colonization capacities and tolerance both to oil pollution and its derivatives (*i.e.*, PAHs).

Sediment acidity also strongly influenced organism distribution. Extremely low pH values were found in CES (average pH—5.81 ± 0.02), which were approximately 1.45 times lower than those observed in the GES (average pH 8.40 ± 0.04) and TES (average pH—8.39 ± 0.06) estuaries. The difference regarding estuary acidity can be directly linked to the individual processes and dynamics of each environment, such as sewage dumping and the decomposition of the organic matter present at each location. Wastewater from sewage or organic matter enrichment, stimulates microbial activity and respiration at different proportions, causing a decrease in pH (*Gallert & Winter, 2005*).

Additionally, previous studies have shown that most industrial effluents and sewage that reach estuarine waters have very low pH values (*Wallace et al., 2014*), which tends to change the pH of these environments. However, groups such as Nematoda and Polychaeta were apparently not negatively affected by the increased acidification in the CES and the high values of organic matter observed at this location. On the other hand, Tardigrada and Gastrotricha were more abundant in GES, which had a more alkaline pH and significantly lower organic matter values, compared to the other estuaries.

The effect of pollutants, especially PAHs, on meiofauna has been previously shown. Studies using meiobenthos are pivotal, both financially and scientifically. Such analyzes are usually cheaper than other biological assessment tools, since meiofauna are highly abundant in the environment, can be handled easily and do not require much storage space in laboratories (*Schratzberger & Ingels, 2018*; *Balsamo et al., 2012*). Scientifically, the use of indices such as EcoQ, are extremely simple and demonstrate satisfactory results when comparing impacted and non-impacted areas (*Danovaro et al., 2004*; *Semprucci, Balsamo & Sandulli, 2016*). However, public policies rarely use the diversity of intra-sedimentary microscopic benthos as ecological tools. This study shows that not only the presence/absence of some taxa in environments with high levels of PAHs can vary, but also that richness (EcoQ) proves to be a useful tool for quick and efficient access to ecological quality. Moreover evenness should serve as an assumption to identify the reliability of the EcoQ test.

## CONCLUSION

We found that PAH contamination followed the urbanization gradient in estuaries with five compounds with sediment concentrations higher than the Fauna Impact Threshold (TEL) in the most urbanized estuary. The reduction in diversity indices, as well as EcoQ (richness), can be used as a good biomonitoring tool for marine conservation, especially

regarding PAHs and it can provide important information on the environmental quality status of the studied estuaries. Such compounds are potentially involved in biodiversity loss within estuaries, in addition to jeopardizing equitability. On the other hand, PAH contamination showed a positive correlation with an increase in organism density, due to the facilitation of the emergence of opportunistic groups like Nematoda and Polychaeta. Furthermore, the autochthonous pollution in the most urbanized estuaries was reflected both in the high organic enrichment and in the accentuated acidification. Granulometry, mainly fine grains that were found at the most polluted places, also played an important role in meiofauna distribution.

### Funding

Renan B da Silva was supported by the grant number IBPG-1244-2.05/20 from Fundação de Amparo a Ciência e Tecnologia do Estado de Pernambuco—FACEPE. Giovanni A P Dos Santos, and Jose Roberto Botelho de Souza, were supported by the PROPESQI Notiz number 09/2019 from Federal University of Pernambuco. Débora A. A. França was supported by the BIC grant number 200216516 from PROPESQI. Raianne Amorim Cavalcante was supported by the BIC grant number 200216456 from PROPESQI, and Andre M. Esteves was supported by the grant number 310249/2019-8 from CNPq, Eliete Zanardi-Lamardo was supported by the grant number 311771/2019-0 from CNPq. The funders had no role in study design, data collection and analysis, decision to publish, or preparation of the manuscript.

### Grant Disclosures

The following grant information was disclosed by the authors:
Fundação de Amparo a Ciência e Tecnologia do Estado de Pernambuco – FACEPE: IBPG-1244-2.05/20.
Federal University of Pernambuco: 09/2019.
PROPESQI: 200216516.
PROPESQI: 200216456.
CNPq: 310249/2019-8.
CNPq: 311771/2019-0.

### Competing Interests

The authors declare that they have no competing interests.

### Author Contributions

- Renan B. da Silva conceived and designed the experiments, performed the experiments, analyzed the data, prepared figures and/or tables, authored or reviewed drafts of the article, and approved the final draft.
- Giovanni A. P. Dos Santos conceived and designed the experiments, performed the experiments, analyzed the data, prepared figures and/or tables, authored or reviewed drafts of the article, and approved the final draft.

- Ana Luiza L. de Farias performed the experiments, prepared figures and/or tables, and approved the final draft.
- Débora A. A. França performed the experiments, prepared figures and/or tables, and approved the final draft.
- Raianne Amorim Cavalcante performed the experiments, prepared figures and/or tables, and approved the final draft.
- Eliete Zanardi-Lamardo conceived and designed the experiments, performed the experiments, analyzed the data, authored or reviewed drafts of the article, analytical Analises of PAH's, and approved the final draft.
- Jose Roberto Botelho de Souza conceived and designed the experiments, performed the experiments, analyzed the data, authored or reviewed drafts of the article, sampling design, and approved the final draft.
- Andre M. Esteves analyzed the data, prepared figures and/or tables, authored or reviewed drafts of the article, and approved the final draft.

### Data Availability

The raw data is available in the Supplemental File.

### Supplemental Information

Supplemental information for this article can be found online at http://dx.doi.org/10.7717/peerj.14407#supplemental-information.

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
