# Peer review of "Effects of PAHs on meiofauna from three estuaries with different levels of urbanization in the South Atlantic"

_PeerJ, doi:10.7717/peerj.14407_

## Round 0.1 · original submission · Major Revisions

Dear Dr. Giovanni Santos

Thank you for the interesting paper and you will find attached three reviews of your work submitted to PeerJ. I follow the reviewer's comments and suggest careful revision of the text, including a detailed description of methods and statistical procedures. Please also try to address comments concerning the use of correlation analysis to address causes of the observed changes in meiofaunal assemblages.

I look forward to receiving your revised submission and a detailed response to the reviewers' concerns

Best regards

Reviewer 1 ·

Basic reporting

The present manuscript is interesting, methods are adeguate and english generally clear. However, authors should improve abstract, introduction and method sections. In this latter, they should give more details on the three study areas: more info on the possible types of disturbances and more info on the PAH origin in the study area section. The discussion can be revised avoiding verbosity . See attached pdf for detailed comments and suggestions

Experimental design

It seems well structured, but further details can be given (see attached pdf)

Validity of the findings

the study is absolutely interesting and will represent an advance in the knowledge on the ecology of benthic fauna and transitional areas

Annotated reviews are not available for download in order to protect the identity of reviewers who chose to remain anonymous.

Reviewer 2 ·

Basic reporting

no comment

Experimental design

Methods ARE NOT described with sufficient detail & information to replicate.

The Data Analysis section is very poor and incomplete. The authors show the results of several analyses, not mentioned in the methods (e.g. Permdisp, Spearman, DistLM, RDA, distance metric...).

There is no references in Lines 202-205

Validity of the findings

Mr. Weber and colleagues described a correlational study to test the hypothesis that (i) environmental factors and (ii) polycyclic aromatic hydrocarbon concentrations significantly correlate with the spatial variation of: density; richness and diversity of meiofauna. Although their data supports the hypothesis, the conclusions they draw from it are very ambitious. The authors must reflect carefully if the correlations founded is enough to establish a causal relationship between PAHs and meiofauna. For example, in lines 557 to 558, the authors states that Tardigrada and Gastrotricha showed greater sensitivity to PAH impact and, in contrast, Nematoda and Polychaeta were more resistant. However, these conclusions are based solely on Correlation analysis (not showed). Another example (lines 550-5510). This study shows that not only do some taxa clearly disappear in the presence of PAHs (actually, the study show only a negative correlation). Correlation does not imply in casualty. It is only a measure of how well the observed patterns of difference among sampling units correlate with each other. Therefore, to say that PAH significantly impact... sounds as the authors conducted a controlled experiment, where the concentration of PAHs was manipulated (holding other environmental factors constant). I suggest reviewing the entire article and avoiding such strong statements. Not less important, Authors should at least recognize possible alternative explanation for the findings, and that observational studies and correlation does not necessarily mean causality, and further studies involving experimental and manipulative approaches are needed.

·

Basic reporting

The paper from da Slva et al. investigates how PAHs, grain size and other major estuary features may affect the meiofauna population. Overall, the topic is relevant to researches devoted to the use of meiofauna as valuable proxy to assess the environmenat quality.

However, the paper need to be improved upon before publication on PeerJ.
- The English need a serious revision, I did some minor corrections (please find the pdf in attachment) but in my opinion it should be revised by an English expert.
- Some references are missed in the introduction section
- Please carefully check all tables and tables in the ESM: there isn't always a correct match between what you mentioned as ESM and table - please check my notes in the pdf.
- Even if the hypotheses were clear and they were confrmed by results, I found along the MS lots of repetions in concepts, e.g. on the importance of meiofauna as biomonitoring tool. Please, be clear and concise.

Experimental design

The experimental design is clear and methods used are well explained.
Research questions are well defined and the sampling design is rigorous and adequate to the research questions addressed by the authors.

Validity of the findings

This study provides interesting findings related to a generally understudied system such as the estuary.

Coclusions can be improved - please find my detailled comment in the pdf. Even if I believe in the use of meiofauna as monitorng tool, the use and interpretation of EcoQ and diversity indices have some limits and results should be always considered not as the 'absolute truth', but as indications of impact (or not).

---

## Round 0.2 · Minor Revisions

Dear Dr. Santos

I consider that your revision addressed most reviewers comments with success. However, I have personally made minor edits and suggestions to your manuscript text in an attempt to make the text clearer and correcting a few typos. Please see the attached pdf file for your reference of these suggestions and return your revisions. In addition, I have a few other comments that you may want to pay attention to:


Figures

Figure 2 - It is unclear if the panel "E" is an overall composition of all estuaries - in that case, as you say in the text that the estuaries have distinct dominances, I suggest removing it from this figure and creating a separate one to illustrate each meiofaunal composition of each estuary separately (this new figure could go to Supplementary files). Alternatively, you could replace Figure 3 with this figure, which may be more informative since the estuaries have distinct abundances.

Figure 3 - I do not understand why plotting the density of the least abundant groups, as this figure is explained in one sentence of your results section. Would it be more useful a figure with overall assemblage composition?

Figures 4,5,6 - I do not understand why run separate MDS and Db plots with distinct variables. You should run one DbRDA analysis with all sediment variables and test for significant correlations. In fact, you have PAH in your Fig 5, so the test presented in Fig 6, although with more detail is repetitive and may be unnecessary since all PAH types point to the most polluted site. Alternatively, if you are confident on the distinct effect of specific PAH types, consider moving your results table (Suppl A8 to the main text). So please consider changing or excluding repetitive DbRDA plots.

Discussion

Lines 529-545 - this is an introductory paragraph, not discussion. consider revising or removing

Your section "Environmental factors and their effects on meiofauna distribution and diversity indices in estuaries with different urbanization levels" is a lot similar to the one above "Environmental factors shaping the distribution of meiofauna and diversity in estuaries with different levels of urbanization". There is as lot of repetition on the effects of sediments and water on the distribution of organisms in both sections. Please consider merging them into one concise and objective section of the discussion.


Thanks for revising your work and I look forward in receiving a revised manuscript

---

## Round 0.3 · accepted · Accept

Thank you for addressing the minor review and for improving the text quality. I have now revised your submission and consider it an excellent contribution to PeerJ. Congratulations on this work.